# Transfer of a Multiclass Method for over 60 Antibiotics in Food from High Resolution to Low Resolution Mass Spectrometry

**DOI:** 10.3390/molecules24162935

**Published:** 2019-08-13

**Authors:** Danilo Giusepponi, Fabiola Paoletti, Carolina Barola, Simone Moretti, Giorgio Saluti, Federica Ianni, Roccaldo Sardella, Roberta Galarini

**Affiliations:** 1Istituto Zooprofilattico Sperimentale dell’Umbria e delle Marche “Togo Rosati”, 06126 Perugia, Italy; 2Department of Pharmaceutical Sciences, University of Perugia, 06123 Perugia, Italy

**Keywords:** antibiotics, liquid chromatography mass spectrometry, milk, muscle, validation

## Abstract

A multiclass method has been developed to screen and confirm a wide range of anti-microbial residues in muscle and milk, and validated using liquid-chromatography coupled to (low-resolution, LR) tandem mass spectrometry (LC-QqQ). Over sixty antibiotics, belonging to ten distinct families, were included in the method scope. The development process was rapidly concluded as a result of two previously implemented methods. This consisted of identical sample treatments, followed by liquid chromatography, and coupled with high-resolution (HR) mass spectrometry (LC-Q-Orbitrap). The validation study was performed in the range between 10–1500 μg·kg^−1^ for muscles and 2–333 μg·kg^−1^ for milk. The main performance characteristics were estimated and, then, compared to those previously obtained with HR technique. The validity of the method transfer was ascertained also through inter-laboratory studies.

## 1. Introduction

Antibiotics are widely used in livestock breeding to treat several diseases that appear in all the food producing animal species. To guarantee public health protection, the European Union requires member states to implement yearly monitoring plans to control the presence of antibiotic residues in food. Therefore, surveillance should be aimed particularly at controlling compliance with the maximum residue limits (MRLs), fixed in Table 1 of the Annex of Regulation (EC) No 37/2010 [1]. For several antibiotics, MRLs have been set in various matrices, such as eggs, fat, honey, kidney, liver, milk, and muscles and still, today, new MRLs are being fixed. In the early 2000s, the liquid chromatography coupled to tandem mass spectrometry technique (LC-QqQ) became essential in the routine analysis of single class of veterinary drug residues in food. Indeed triple quadrupole mass spectrometry analyzers were able to assure both greater sensitivity and selectivity than the traditional LC detectors, based on UV-Vis and fluorescence spectroscopy. In addition, for some important classes, such as aminoglycosides or avermectins, the need of a derivation step could be avoided. In the last ten years, the improvement of LC-QqQ systems allowed the realization of a further step in drug residue analysis, introducing procedures that are able to determine simultaneously more than one drug class [2,3,4]. As consequence, a remarkable effort has been made to progressively replace single-class with multiclass protocols, since this is a cost-effective way to improve the current residue control programs, thereby ensuring the determination of a wide number of compounds, with only few methods. Reviewing the main relevant published papers, some research groups recurred (Table 1). Among the control laboratories, the Official Food Control Authority of Zurich (Zurich, Switzerland), the RIKILT (Wageningen, Netherlands), the European Union Reference Laboratory for Antimicrobial Residues in Food (EURL, Fougères, France), the National Institute for Agrarian and Veterinary Research (INIAV, Vila do Conde, Portugal), the Istituto Zooprofilattico Sperimentale dell’Umbria e delle Marche (IZSUM, Perugia, Italy), the Canadian Food Inspection Agency (Calgary, Canada), the Residue Analysis Laboratory of Laboratório Nacional Agropecuário (LANAGRO, Porto Alegre, Brazil), and the US Department of Agriculture (USDA, Wyndmoor, PA, USA) are mentioning.

The universities of Barcelona (Spain), Almeria (Spain), and Athens (Greece) have been the most active in this analytical field. LC-QqQ techniques are the most consolidated and most common multiclass procedures for veterinary drugs. These techniques have been mainly developed using this type of equipment [5,6,10,11,12,16,17,19,20,23,27]. In 2008–2009, the Official Food Control Authority of Zurich and the Dutch RIKILT Institute proposed, for the first time, the application of high-resolution (HR) mass spectrometry, based on time-of-flight (TOF) technology [7,8,9]. About three years later, the same Laboratory of Zurich, and the research group of Almeria University developed multiclass procedures for veterinary drugs, respectively, in meat, and milk, using LC-Orbitrap technique, a new MS analyzer, that was commercialized in 2005 [13,29]. Later, the introduction of benchtop hybrid high-resolution mass spectrometers (mainly, Q-TOF and Q-Orbitrap) produced further advantages in terms of selectivity and accuracy and, accordingly, these kinds of equipment has been more commonly applied (Table 1) [14,15,18,21,22,24,25,26,28,29].

Based on all the above, multiclass methods are no longer innovative procedures, and there is interest in their wide diffusion. The possibility of easy implementation and sustainable daily management, independent from the available LC-MS equipment. The aim of this work was to discuss the transfer of previously developed multiclass methods for more than sixty antibiotics in meat and milk from an LC-Q-Orbitrap platform to an LC-QqQ one [24,25]. The performance characteristics of the new LC-QqQ methods were estimated by means of full validation studies carried out according to European Commission Decision 2002/657/EC [30]. Finally, a comparison between the two techniques was carried out in the light of their cost-effectiveness in routine analysis of veterinary drug residues.

## 2. Results and Discussion

### 2.1. Optimization of LC-MS/MS Conditions

The choice of analytes has been carried out using the most administered antibiotics in farm. Only the classes of aminoglycosides and colistins were excluded, as their high polarity hampers the chromatographic retention, based on the reversed-phase mechanism (C18 column). On the other hand, the addition of ion-pairing agents on the mobile phase produced remarkable ion suppression, with detrimental effects on all the other analytes [24]. The chromatographic conditions were optimized starting from the parameters set for the LC-Q-Orbitrap methods. In order to profitably increase analyte retention, the percentage of methanol (eluent B) was reduced from 5% down to 2% (by volume). According to a typical reversed-phase mechanism, this change allowed us to obtain retention times of about 0.5 min higher than the initial tested conditions (Appendix A). 

The MS conditions were established without the infusion of the individual solutions of analytes, but by setting the transitions on the basis of the ion fragments previously studied [24]. As shown in Table 2, apart few exceptions, such as some beta-lactams ([M + Na]^+^), sulfanilamide ([M + H − NH_3_]^+^), spiramycin, neospiramycin, cefquinome, tildipirosin, tilmicosin, tulathromycin marker, and tulathromycin ([M + 2H]^++^), the selected precursor ion species were generally the protonated molecular ions ([M + H]^+^). For macrolides, it is not uncommon for the choice of bi-charged ions to be used as a precursor, due to their favorable abundance among the formed charged species [31]. The sample preparation was exactly the same as that previously optimized by Moretti et al. [24,25]; however, two internal standards (ISs) were replaced, in order to either, decrease costs (metacycline instead of tetracycline-d6), or to improve the MS response (ceftiofur-d3 instead of cefadroxil-d4). In this context, the ISs were not used for quantification purposes, but only to perform the internal quality control by checking the success of the analytical operations, during the routine application of the procedure as well as to monitor the run-to-run differences in the retention times [8]. For this purpose, at the beginning of sample treatment, IS were added at 10 µg·kg^−1^ and, before the release of the results, the presence (S/N > 3) of all eight compounds must be verified. The analyte quantification was achieved by matrix-matched curves (external standardization), which corrected the concentration for the relevant recovery factor [32]. The LC-QqQ chromatograms of a blank muscle, and of the same spiked at 10 µg·kg^−1^, are reported in Figure 1 and Figure 2, respectively. Eight representative analytes are shown, starting from the polar metabolite of florfenicol (florfenicol amine, RT = 3.4 min) to the last eluting compound (rifaximin, RT = 20.7 min). The analogous chromatograms are shown also for milk (Figure 3 and Figure 4).

### 2.2. Method Validation

Selectivity requirements are reported in Commission Decision 2002/657/EC [30]. The ion ratio of the two selected transitions (Table 2), and their relative retention times (<2.5%), were checked to confirm analyte identification. Linearity in the matrix was evaluated with five-points matrix-matched curves: 2, 10, 33, 100, and 150 µg·kg^−1^. Therefore, levels higher than 150 µg·kg^−1^ had to be tested, and the final extract was diluted ten-fold or more, as reported in Appendix A. The linearity data are summarized in Appendix A. For several analytes, the first calibration point (2 µg·kg^−1^) had to be discarded, due to the scarce response. In other more critical cases (e.g., cefacetrile in meat/muscle, tildipirosin and tulathromycin markers in milk) additional points have been removed. Since Commission Decision 657/2002/EC [30] does not furnish precise criteria for evaluating linearity, the “Guidance document on analytical quality control and validation procedures for pesticide residues analysis in food and feed” was followed [33]. The percentage deviation of the back-calculated concentrations (C_measured_) from the true concentrations (C_true_) was calculated (1):(1)$$Deviation\;(\%)=\frac{\left(C_{measured}-C_{true}\right)}{C_{true}}\cdot 100$$

For each calibration point, it was verified that its value was not more than ± 20%. As an example, the calibration data of six analytes, belonging to different antibiotic families, are reported in Appendix A and Appendix A.

The overall recoveries and precisions data are listed in Table 3. For meat, seven validation levels were performed in the range 10–1500 µg·kg^−1^, whereas, in milk, five levels were investigated (10–333 µg·kg^−1^), with an additional concentration at 2 µg·kg^−1^ to check amoxicillin, ampicillin, and penicillin G accuracy at ½ MRL, as required by the Commission Decision 2002/657/EC [30]. Moreover, in milk, two additional molecules were tested with respect to the original group of 62 compounds, that is, the two quinolones, nalidixic acid, and norfloxacin.

In this work, the classical validation scheme (0.5 MRL, MRL and 1.5 MRL), described in Commission Decision 2002/657/EC, was not applied. Paragraph 3.1.3. of the same decision allows the introduction of alternative models [30] and, since the validation studies of multiclass procedures have to consider dozens of MRLs, which can vary also in function of the animal species, the adoption of progressive validation levels, that are equal for all the analytes, was fully justified. On the other hand, when the Commission Decision was issued (2002), the development of multiclass procedures for the control of veterinary drug residues was not initiated. The spiking ranges were chosen, considering, on the one side, the reachable concentrations, and, on the other side, the relevant MRLs [1] in all the couplings analyte/matrix. Preliminary experiments demonstrated that, at levels lower than 10 µg·kg^−1^, the precision of several amphenicols, macrolides, beta-lactams, and tetracyclines became unsatisfactory.

The recovery factors (Rec) were established by comparing the peak area of each compound in the spiked samples against the peak area in the matrix matched standards, by which the antibiotics were added immediately prior to LC injection. The data in Table 3 summarize the average recoveries obtained in the whole validation range. In muscles, all recoveries were higher than 70%, except the majority of tetracyclines (62–69%), three beta-lactams (64–68%), and two macrolides (neospiramycin, 67% and tulathromycin, 69%). Analogously, in milk, recoveries higher than 70% were generally obtained, except for the more polar sulfonamides (61–70%) and one macrolide (erythromycin A, 67%). Since the quantification was performed with an external standardization, the raw results were always corrected for the relevant recovery factor, in order to correct the systematic error [32]. With regard to precision, the coefficients of variations (CVs) were calculated at each validation level both, repeatedly, and in intra-lab reproducibility conditions (CV_r_, and CV_wR_, respectively), by applying ANOVA. Moreover the CV_wR_ (and CV_r_) were pooled to obtain an overall precision index, namely CV_wR,pooled_ (2),
(2)$$CV_{wR,pooled}=\sqrt{\frac{(n_{1}-1)CV_{wR1}^{2}+(n_{2}-1)CV_{wR2}^{2}+...(n_{n}-1)CV_{wRn}^{2}}{(n_{1}-1)+(n_{2}-1)+...}}$$ where *CV_wR1_, CV_wR2_ ... CV_wRn_* were the coefficients of variation at the increasing levels 1*, 2 … n; n_1_, n_2_…n* were the number of replicates at each level [34]. For a certain analyte, the *CV_wR,pooled_* give a single estimate of precision, which can be applied to calculate decision limits (CCα) and detection capabilities (CCβ) at whatever MRL value. Therefore, the *CV_wR,pooled_* can also be used to obtain CCα and CCβ, where new MRLs were fixed in Regulation 37/2010 [1], maximizing the cost-effectiveness of the validation study. The decision limit and detection capability were calculated as follows (equations 3 and 4):(3)$$CC\alpha_{MRL}=MRL+1.64\cdot CV_{wR,pooled}\cdot MRL$$
(4)$$CC\beta_{MRL}=CC\alpha_{MRL}+1.64\cdot CV_{wR,pooled}\cdot CC\alpha_{MRL}$$

In Appendix A, the MRLs of the 64 antibiotics, in various food-producing animal species, and in bovine milk, together with the relevant CCαs and CCβs, are listed. The MRLs are those reported in the last consolidated text of Regulation 37/2010 [1,35]. In muscles, valnemulin demonstrated high imprecision (CV_wR,pooled_ = 31%) and, therefore, for these compounds, the developed procedure could be only used for screening purposes (Table 3). For all the other antibiotics, CV_wR,pooled_ was always lower than 20%. On the other hand, in milk, cefacetrile and tulathromycin marker demonstrated insufficient precision (CV_wR,pooled_ > 22%). Matrix effects (ME%) listed in Table 3 were calculated as follows (equation 5),
(5)$$ME(\%)=\frac{b_{MM}}{b_{S}}\times 100$$ where *b*_MM,_ and *b*_S_ were the slopes of matrix matched curves, and solvent standard curves, i.e., curves prepared in ammonium acetate 0.2 M, respectively. In the whole, although the sample purification was scarce, the matrix effects (suppression or enhancement) were limited and very few compounds demonstrated ME (%) higher than |50|%. This was probably because the long chromatographic run (30.5 min) allowed the distribution of matrix-interfering compounds and analytes, from preventing excessive bunching [26].

Since the signal-to-noise approach (S/N) is rather subjective [36], “operative” (fit for purpose) LOD and LOQ were fixed by examining the precision at each validation level (Appendix A). All the compounds were detectable at the first concentration, i.e., 2 µg·kg^−1^ for amoxicillin, ampicillin, and penicillin G in milk and 10 µg·kg^−1^ for all the others. The only exception was cefacetrile, which is a scarcely ionizable molecule and, therefore, detectable and quantifiable from 33 µg·kg^−1^ in muscle (CV_r_ = 13%, CV_wR_ = 14%, recovery = 76%) and only detectable in milk (insufficient precision). A satisfactory accuracy for both amoxicillin and ampicillin was obtained at 2 µg·kg^−1^, whereas penicillin G demonstrated unsatisfactory precision at this level (CV_r_ = 23%, CV_wR_ = 26%) and, therefore, this beta-lactam could be quantified only starting from 10 µg·kg^−1^ (CV_r_ = 7.5%, CV_wR_ = 11%). Therefore, penicillin G, a fundamental drug for the treatment of sub-clinical mastitis in **lactating cows**, could not be quantified in milk at ½ MRL (2 µg·kg^−1^) and the method was suitable only for screening purpose [30]. In Appendix A the LC-QqQ chromatograms of these three compounds are shown in a blank milk sample and in the same sample spiked at 2 µg·kg^−1^.

### 2.3. Comparison of LC-QqQ and LC-Q-Orbitrap Methods

Comparing the recovery factors of the methods developed in meat, similar results were always obtained (differences < 15%), except for cefquinome, which demonstrated a higher recovery when LC-QqQ was applied (+ 26%: 97% LR vs. 71% HR) and valnemulin which, nevertheless, was not accurately quantifiable with the LR procedure, as discussed above. The recovery differences are visualized in Appendix A. Examining the data of the more polar cephalosporins (desacetylcephapirin, cephapirin, cefacetrile and cefalonium), cefquinome recovery appeared suspect (over-estimated) using QqQ technique. Since the sample preparation protocol was the same, this difference should be attributed to an instrumental technique. Interestingly, cefquinome co-eluted with sulfamerazine and, in addition, accidentally their precursor ion had the same nominal mass charge ratio i.e., *m*/*z* 265 (Table 2). In our laboratory, at the beginning of the development of multiclass methods for antibiotics in food (2014), two separate matrix-matched curves were prepared, one for beta-lactams and another for all other analytes [24]. This measure was precautionary in order to avoid the possible negative effects of methanol, contained in the intermediate solution of antibiotics other than beta-lactams (see Section 2.2) on beta-lactam stability, as described in the literature [37,38]. In saying that, the artefacts in the quantification of cefquinome and/or sulfamerazine could occur if “hidden” transitions were shared between these two compounds. This means that a transition, monitored only for one of the two analytes (Table 2), was shared by the other one, too. In order to verify this hypothesis, two individual solutions (50 ng·mL^−1^) were separately injected, by simultaneously monitoring the four relevant SRMs (Appendix A). It was evident that sulfamerazine shared a “hidden” transition with cefquinome (*m*/*z* 265 > *m*/*z* 199—left side of Appendix A), but also viceversa (*m*/*z* 265 > *m*/*z* 156—right side of Appendix A). For sulfamerazine the fragment ion species at *m*/*z* 199 was formed by the rearrangement ion, by losing H_2_SO_2_ from the protonated molecule [39]. On the other hand, for cefquinome, the ion *m*/*z* 156 derived by the cleavage of C-C bond between oxime and the carbonyl group of C^7^ amide ([C_5_H_6_N_3_OS]^+^) of the beta-lactam ring [40]. However, from a quantitative point of view, sulfamerazine could notably affect the peak area of cefquinome, but the contrary was much less, since sulfamerazine responded more significantly than cefquinome (about 4.9 × 10^6^ vs. 1.1 × 10^6^). Accordingly, when sulfamerazine was co-present, cefquinome concentration was significantly over-estimated. Later, observing that, in the adopted experimental conditions, beta-lactams were not deteriorated by methanol, the validation study in milk was performed by preparing only one matrix-matched calibration curve with all the analytes, including beta-lactams. In summary, although the development of LR procedures has been very fast and effective, due to previously-studied conditions [24,25], for one analyte, i.e., cefquinome, the choice of the precursor ion should be re-evaluated.

In Appendix A the differences between LR and HR recovery factors in milk are shown. Tulathromycin marker and cefquinome had better recovery rates using HR detection, whereas valnemulin, cloxacillin, and rifaximin, showed better results using the LR system. As reported in our previous paper [25], LC-Q-Orbitrap suffers from “post interface ion suppression”, which consists of instrument saturation when intense matrix-related compounds are present [41]. This phenomenon was more pronounced for the last eluting compounds, such as valnemulin, cloxacillin, and rifaximin, which explains the observed data in milk extracts. These have more interfering substances with respect to muscle [25]. Comparing the precision data (intra-laboratory reproducibility, CV_wR,pooled_, see Appendix A), remarkable differences (≥15%) between the two techniques were observed only in milk and, again, tulathromycin marker and cefquinome revealed the worse performances when determined by LC-QqQ. C_wR,pooled_ of tylosin, cloxacillin, and tylvalosin were about 10% higher when analyzed by LC-Q-Orbitrap (Appendix A).

According to accreditation rules [42], since 2014 our laboratory participated in Proficiency Test Schemes in meat and milk, by applying the LC-Q-Orbitrap methods in these products. Some of the stored test materials were then re-tested, by applying the new developed procedures. The results, together with the consensus values assigned by the Organizers, are listed in Table 4. Examining the acceptability ranges, satisfactory z-scores would have been always obtained, except in the case of amoxicillin in milk (sample code: MI1532-A1). This latter result was explainable with the well-known instability of penicillin antibiotics [43,44].

In summary, the main advantages and disadvantages of the two techniques are undoubtedly, the QqQ analyser, which only involves only a kind of acquisition, i.e., SRM mode. On the contrary, Q-Orbitrap forces more complex experiments, also because the detection of analytes at trace levels is complicated by the “post interface ion suppression” phenomenon. On this subject, we demonstrated that, in certain chromatographic regions, for milk it is not possible to reach the required limits using full-scan experiments, since the massive presence of interfering substances can drastically worsen the sensitivity [25]. With regard to the sample throughput, the sample preparation protocol is identical and, therefore, there is no great difference. Moreover, it must be highlighted that some performances of LR system are due to the obsolescence of the available equipment. For example, most likely, a more recent LR platform could reach comparable LODs with HR, i.e., lower than 10 µg·kg^−1^. All that said, the LC-QqQ technique is more suitable for routine laboratories, considering its user-friendliness and lesser cost (about three time lesser than LC-Q-Orbitrap).

## 3. Experimental

### 3.1. Chemical and Reagents

Acetonitrile (ACN) and methanol (LC-MS grade) were from Carlo Erba Reagents (Milan, Italy). Formic acid (50%) and *N*,*N’*-dimethylformamide (DMF) were purchased from Sigma-Aldrich (St. Louis, MO, USA). EDTA sodium salt dehydrate and ammonium acetate were provided by Sigma-Aldrich. Ultra-pure deionized water was generated by a Milli-Q purification apparatus (Millipore, Bedford, MA, USA). Amoxicillin, ampicillin, cloxacillin, dicloxacillin, nafcillin, oxacillin, penicillin G (benzylpenicillin), penicillin G-d7, penicillin V (phenoxymethylpenicillin), cefalonium, cefoperazone, cefquinome, ceftiofur, cephalexin, ciprofloxacin, danofloxacin, difloxacin, enrofloxacin, flumequine, marbofloxacin, nalidixic acid, norfloxacin, oxolinic acid, sarafloxacin, erithromycin A, spiramycin I, tylosin A, tilmicosin, sulfadiazine, sulfaguanidine, sulfadimethoxine, sulfamerazine, sulfamethazine (sulfadimidine), sulfamethoxazole, sulfanilamide, sulfapyridine, sulfaquinoxaline, sulfathiazole, trimethoprim, chlortetracycline, doxycycline, metacycline, oxytetracycline, tetracycline, florfenicol, florfenicol amine, thiamphenicol, lincomycin, rifaximin, tiamulin and valnemulin were obtained from Sigma-Aldrich. Sulfamonomethoxine was purchased from Dr. Ehrenstorfer (Augsburg, Germany); cefazolin, cefacetrile, ceftiofur-d3, cephapirin, desacetylcephapirin, florfenicol-d3, 3-*O*-acetyltylosin, gamithromycin, neospiramycin, spiramycin I-d3, tildipirosin, tulathromycin, tylvalosin, 4-epi-chlortetracycline, 4-epi-tetracycline, 4-epi-oxytetracycline, and tulathromycin marker (CP-60,300) were purchased from TRC Inc. (Toronto, Canada); sulfamethazine-^13^C6, sulfanilamide-^13^C6 and enrofloxacin-d5 were obtained from WITEGA (Berlin, Germany).

### 3.2. Standard Solutions

Individual stock standard solutions of 100 µg·mL^−1^ were prepared with methanol (amphenicols, lincosamides, macrolides, pleuromutilins, sulphonamides, tetracyclines and trimethoprim). The solubilization and storage conditions were previously studied [24]. Beta lactams were solubilized in H_2_O/ACN 75/25 (*v*/*v*), except ceftiofur (DMF). Quinolones in MeOH/H_2_O 80/20 (*v*/*v*), except ciprofloxacin, nalidixic acid, norfloxacin, enrofloxacin and oxolinic acid (DMF). Rifaximin was prepared in MeOH/H_2_O 50/50 (*v*/*v*). The stock solutions were stored at −20 °C with variable storage times: From 1 month (cefquinome) to 24 months (sulphonamides). Working solutions (10, 1 and 0.1 µg·mL^−1^) were prepared from the relevant stock solutions diluting with H_2_O/ACN 75/25 (*v*/*v*) for beta-lactams and with methanol for all the other antibiotics. The solutions of internal standards were prepared according to their native compound or class.

### 3.3. LC-MS-MS Conditions

LC-MS/MS measurements were performed by a Surveyor LC pump, coupled with a triple quadrupole mass spectrometer (TSQ Quantum Ultra, Thermo Fisher, San Jose, CA, USA), an electrospray source included, and operating in a positive ionization mode. Separation was achieved on a Poroshell 120 EC-C18 column (3.0 × 100 mm; 2.7 μm particle diameter), which was connected to a guard column Poroshell (2.1 × 5 mm), both from Agilent Technologies (Santa Clara, CA, USA). The flow rate used was 0.25 mL·min^−1^ and the column temperature set at 30 °C. Mobile phase A was an aqueous solution 0.1% (*v*/*v*) formic acid and eluent B methanol. The gradient profile started at 2% eluent B for 1 min and increased linearly up to 95% B in 19.5 min; this condition was maintained for 5 min before returning to initial condition in 1 min (2% B) and held for 4 min to equilibrate the column. The sample temperature was kept at 16 °C and the injection volume 10 µL. For MS detection, the parameters were the follows: Capillary temperature 300 °C, vaporizer 320 °C, spray voltage 3 kV, and a resolution setting of Q1 and Q3 *m*/*z* 0.7. Sheath gas and auxiliary gas (nitrogen) pressures were set at 35, and 15 arbitrary units, respectively. Collision gas (argon) pressure: 1.5 mtorr. The collision energies, that were associated with each transition, are listed in Table 2.

### 3.4. Sample Preparation 

The sample preparation was described elsewhere [24,25]. Briefly, (1.50 ± 0.01) g of minced muscle or milk were weighed in a 50 mL Falcon tube. The sample was spiked with: (i) 15 µL of the solution of the two internal standards (ISs) of beta-lactams at 1 µg·mL^−1^ (ceftiofur-d3 and penicillin G-d7); (ii) 15 µL of the solution of the other six ISs (sulfanilamide-^13^C6, sulfamethazine-^13^C6, enrofloxacin-d5, florfenicol-d3, spiramycin-d3 and metacycline) at 1 µg mL^−1^. For muscles, one hundred microliters of 0.1 M of EDTA was then added, and the sample was extracted with 3 mL of a mixture of acetonitrile/water 80/20 (*v*/*v*). Milk was extracted with one milliliter of 0.1 M of EDTA and 3 mL of acetonitrile. A second extraction with 3 mL of pure acetonitrile was performed for both matrices. After centrifugation, the reunited extracts were evaporated and solubilized in 1.5 mL of ammonium acetate 0.2 M. Ten µL was injected in the LC system. 

### 3.5. Method Validation

The validation was carried out following the Commission Decision 2002/657/EC [30]. To test the selectivity, blank muscle samples, belonging to the main animal species (bovine, swine and poultry) and bovine milk samples from different origins, were analyzed. The linearity in the matrix has been evaluated in the range 2–150 µg·kg^−1^ (2, 10, 33, 100 and 150 µg·kg^−1^). The matrix-matched solutions were prepared by adding the analytes immediately prior to LC injection. The curves were constructed, including at zero (blank), and by plotting the average peak area of analyte (three injections for each concentration point) against its concentration. An unweighted linear regression model was applied to the calibration data. The precision (repeatability and within-laboratory reproducibility), recovery (trueness), decision limit (CCα), and detection capability (CCβ) were studied following the experimental plans, described in Appendix A (muscle) and Appendix A (milk) of the Appendix A. A blank bovine muscle or milk was spiked at the beginning of the extraction procedure with the appropriate standard solutions. Four replicates (*n* = 4) at each level were carried out on the same day, along with the relevant matrix-matched calibration curve. Each series was repeated on three different days, and at varying times, operator, and calibration status of the LC-MS system. The spiking levels in muscles were; 10, 33, 100, 150, 333, 1000, and 1500 µg·kg^−1^ and 10, 33, 100, 150, and 333 µg kg^−1^ in milk. In milk, penicillin G, amoxicillin, and ampicillin were also tested at 2 µg·kg^−1^. The Limits of Detection (LODs ) and Quantification (LOQs) were estimated on the basis of the observed accuracy (recovery and precision) at the first validation concentration or, if necessary, at the second one.

## 4. Conclusions

The validity of the transfer, from LC-Q-Orbitrap to LC-QqQ of two multiclass methods for veterinary drugs in food, has been demonstrated. Using the LR platform, valnemulin in muscle, cefacetrile, and tulathromycin marker in milk did not reach acceptable precision. In addition, examining LC-QqQ results, a series of accidental events (co-elution, selected precursor ions, fragmentation pathway etc) produced an over-estimation of cefquinome in meat, due to the co-presence of sulfamerazine. If neglected, this phenomenon could give false positive results. Since the quantification process of the LC-Q-Orbitrap system was based on a different principle (peak area of the precursor ion measured with MS accuracy < 5 ppm), this drawback was not observed. However, in milk, LC-Q-Orbitrap achieved worse recoveries for some of the more non-polar analytes, eluting in the chromatographic zone in which the interfering substances were more abundant. The “post interface ion suppression”, which is a specific phenomenon of Orbitrap mass analysers could explain this latter evidence. A further consideration concerns the chromatographic separation, which is fundamental when highly selective detectors are applied. A good separation of peaks, avoiding the overlapping and bunching of both analytes and endogenous substances, minimizes the risk of false positive results and reduces matrix effects. Finally, the described method transfer has been successfully performed on an obsolete LC-QqQ platform (fifteen year-old), encouraging the implementation of multiclass strategy also in routine laboratories with limited instrumental resources.

## Figures and Tables

**Figure 1 molecules-24-02935-f001:**
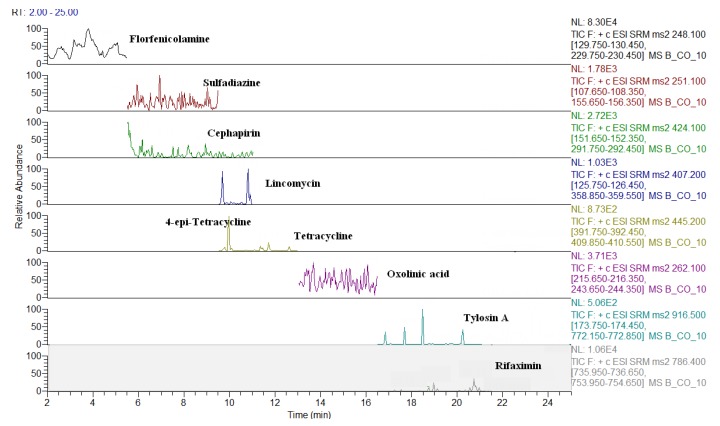
LC-QqQ chromatograms of a blank bovine muscle.

**Figure 2 molecules-24-02935-f002:**
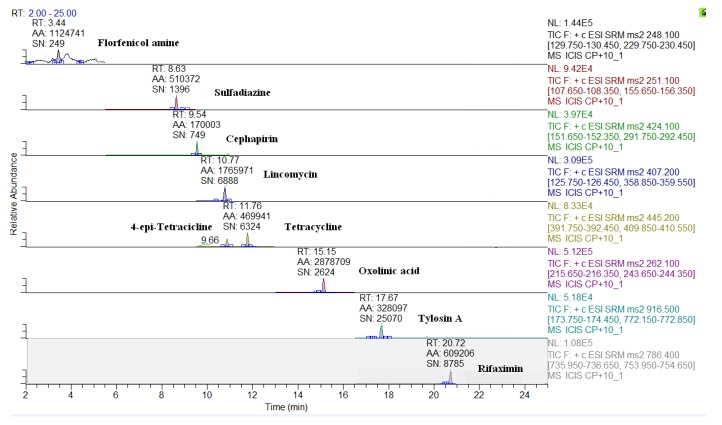
LC-QqQ chromatograms of a spiked bovine muscle (10 µg·kg^−1^).

**Figure 3 molecules-24-02935-f003:**
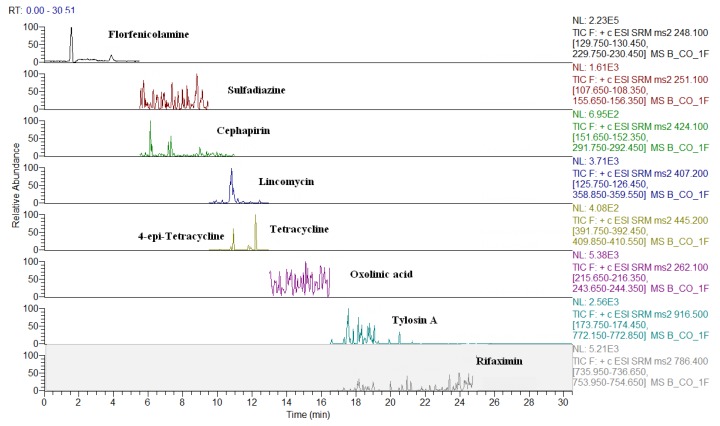
LC-QqQ chromatograms of a blank bovine milk.

**Figure 4 molecules-24-02935-f004:**
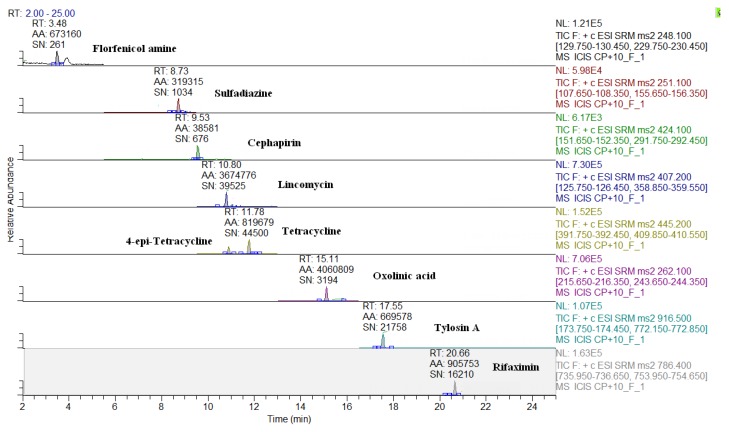
LC-QqQ chromatograms of a spiked bovine milk (10 µg·kg^−1^).

**Table 1 molecules-24-02935-t001:** Overview of multiclass methods for the determination of veterinary drug residues in tissues and milk.

	N° of Veterinary Drugs	Matrix	Equipment	Reference	Laboratory/Centre ^a^
1	18	Milk	LC-QqQ	Aguilera-Luiz et al. 2008 [5]	Almeria University (Spain)
2	39	Chicken muscle	LC-QqQ	Chico et al. 2008 [6]	Barcelona University (Spain)
3	>100	Muscle	LC-TOF	Kaufmann et al. 2008 [7]	OFCA-Zurich (Switerland)
4	>100	Milk	LC-TOF	Stolker et al. 2008 [8]	RIKILT (The Netherlands)
5	Ca 100	Meat and other food	LC-TOF	Peters et al. 2009 [9]	RIKILT (The Netherlands)
6	Ca 26	Animal tissues	LC-QqQ	Stubbings et al. 2009 [10]	FERA (UK)
7	58	Milk	LC-QqQ	Gaugain-Juhel et al. 2009 [11]	EURL (France)
8	21	Milk	LC-QqQ	Martinez-Vidal et al. 2010 [12]	Almeria University (Spain)
9	30	Milk	LC-Orbitrap, LC-Q-TOF, LC-QqQ	Romero-González et al. 2011 [5]	Almeria University (Spain)
10	>100	Meat and other food	LC-Orbitrap	Kaufmann et al. 2011 [13]	OFCA-Zurich (Switzerland)
11	>60	Meat	LC-LTQ-Orbitrap	Hurtaud-Pessel et al. 2011 [14]	EURL (France)
12	59	Milk and honey	LC-Q-TOF	Wang et al. 2012 [15]	CFIA-Calgary (Canada)
13	21	Meat	LC-QqQ	Bittencourt et al. 2012 [16]	LANAGRO (Brazil)
14	24	Milk and liver	LC-QqQ	Martins et al. 2014 [17]	LANAGRO (Brazil)
15	>100	Milk	LC-Q-Orbitrap	Kaufmann et al. 2014 [18]	OFCA-Zurich (Switzerland)
16	39	Liver	LC-QqQ	Freitas et al. 2015 [19]	INIAV (Portugal)
17	23	Liver	LC-QqQ	Martins et al. 2015 [20]	LANAGRO (Brazil)
18	>100	Milk	LC-Q-Orbitrap	Wang et al. 2015 [21]	CFIA-Calgary (Canada)
19	>100	Various food	LC-Q-TOF	Dasenaki et al. 2015 [22]	University of Athens (Greece)
20	76	Bovine muscle	LC-QqQ	Dasenaki et al. 2016 [23]	University of Athens (Greece)
21	62	Animal muscle	LC-Q-Orbitrap	Moretti et al. 2016 [24]	IZSUM (Italy)
22	62	Milk	LC-Q-Orbitrap	Moretti et al. 2016 [25]	IZSUM (Italy)
23	>120	Animal tissues	LC-QqQ/LC-Q-TOF	Anumol et al. 2017 [26]	USDA (USA)
24	174	Bovine tissues	LC-QqQ	Lehotay et al. 2018 [27]	USDA (USA)
25	44	Salmon	LC-Q-TOF	Gaspar et al. 2019 [28]	INIAV (Portugal)

^a^ OFCA = Official Food Control Authority; CFIA:Canada Food Inspection Agency; FERA: The Food and Environment Research Agency; LANAGRO: Laboratório Nacional Agropecuário; INIAV: Instituto Nacional de Investigação Agrária e Veterinária; USDA: United States Department of Agriculture; IZSUM: Istituto Zooprofilattico Sperimentale dell’Umbria e delle Marche.

**Table 2 molecules-24-02935-t002:** Summary of the selected reactions transitions (SRM) monitored for the sixty-four targeted analytes.

N°	Analyte	Retention Time (min)	Adduct (*m*/*z*)	Precursor Ion (*m*/*z*)	Product Ions (*m*/*z*)	Collision Energy (eV)
1	Sulfaguanidine	2.85	[M + H]^+^	215.1	92.0	15
					156.0	20
2	Florfenicolamine	3.20	[M + H]^+^	248.1	230.1	10
					130.1	30
3	Sulfanilamide	3.30	[M + H − NH_3_]^+^	156.0	92.0	12
					108.1	10
	Sulfanilamide-13C6	3.30	[M + H − NH_3_]^+^	162.0	98.1	13
					114.1	13
4	Desacetylcephapyrin	6.80	[M + H]^+^	382.1	152.0	30
					226.0	20
5	Amoxicillin	8.30	[M + H]^+^	366.1	349.1	10
					114.0	20
6	Sulfadiazine	8.50	[M + H]^+^	251.1	108.0	26
					156.0	15
7	Sulfathiazole	9.20	[M + H]^+^	256.0	92.1	28
					156.0	15
8	Cephapyrin	9.45	[M + H]^+^	424.1	292.1	20
					152.0	30
9	Sulfapyridine	9.50	[M + H]^+^	250.1	108.0	26
					156.0	17
10	Tildipirosin	9.90	[M+2H]^++^	367.7	281.2	20
					98.1	18
11	Sulfamerazine	9.90	[M + H]^+^	265.1	108.0	27
					156.0	17
12	Cefquinome	10.00	[M + 2H]^++^	265.1	134.2	20
					199.1	20
13	Cefacetrile	10.15	[M + Na]^+^	362.0	258.0	10
					302.0	10
14	Cefalonium	10.50	[M + H]^+^	459.1	337.0	10
					152.0	20
15	Lincomycin	10.50	[M + H]^+^	407.2	126.1	30
					359.2	10
16	Tulathromycin marker	10.60	[M + 2H]^++^	289.0	158.3	17
					420.5	17
17	Thiamphenicol	10.60	[M + H]^+^	356.0	308.0	20
					229.0	20
18	Epitetracycline	10.60	[M + H]^+^	445.2	410.2	20
					392.1	30
19	Trimethoprim	10.70	[M + H]^+^	291.1	261.1	30
					230.1	30
20	Marbofloxacin	10.80	[M + H]^+^	363.1	276.1	14
					320.1	14
21	Sulfamethazine	11.10	[M + H]^+^	279.1	92.1	31
					124.1	28
	Sulfamethazine-^13^C6	11.10	[M + H]^+^	285.1	186.1	17
22	Epioxytetracycline	11.35	[M + H]^+^	461.2	426.1	20
					337.1	30
23	Norfloxacin^a^	11.50	[M + H]^+^	320.1	231.2	39
					282.1	29
24	Tetracycline	11.50	[M + H]^+^	445.2	410.2	20
					392.1	30
25	Cefalexin	11.70	[M + H]^+^	348.1	158.0	10
					174.1	20
26	Oxytetracycline	11.80	[M + H]^+^	461.2	426.1	20
					337.1	30
27	Ciprofloxacin	11.80	[M + H]^+^	332.1	245.1	23
					288.1	17
28	Enrofloxacin	11.90	[M + H]^+^	360.2	245.0	26
					316.1	19
	Enrofloxacin -d5	11.90	[M + H]^+^	365.2	321.4	18
29	Tulathromycin	11.90	[M + 2H]^++^	404.0	158.1	20
					116.1	20
30	Danofloxacin	11.95	[M + H]^+^	358.2	283.1	24
					340.1	22
31	Cefazolin	12.00	[M + H]^+^	455.0	323.1	10
					156.0	20
32	Sulfamethoxazole	12.10	[M + H]^+^	254.1	108.1	28
					156.0	16
33	Difloxacin	12.30	[M + H]^+^	400.1	299.1	28
					356.1	18
34	Ampicillin	12.30	[M + H]^+^	350.1	106.1	20
					160.0	20
35	Sulfamonomethoxine	12.30	[M + H]^+^	281.1	108.1	28
					156.0	16
36	Florfenicol	12.40	[M + H]^+^	358.0	241.0	20
					340.0	10
	Florfenicol -d3	12.40	[M + H]^+^	361.0	241.0	16
37	Cefoperazone	12.60	[M + H]^+^	646.1	530.3	10
					143.1	30
38	Sarafloxacin	12.60	[M + H]^+^	386.1	342.1	18
					299.1	26
39	Epichlortetracycline	12.85	[M + H]^+^	479.1	444.1	20
					154.0	30
40	Neospiramycin	13.40	[M + 2H]^++^	350.2	160.1	10
					174.1	20
41	Chlortetracycline	13.80	[M + H]^+^	479.1	441.1	20
					154.0	30
42	Spiramycin	14.05	[M + 2H]^++^	422.3	702.4	10
					174.1	20
	Spiramycin -d3	14.05	[M + 2H]^++^	423.8	174.0	20
43	Sulfadimethoxine	14.40	[M + H]^+^	311.1	108.1	29
					156.0	21
44	Sulfaquinoxaline	14.80	[M + H]^+^	301.1	92.1	30
					156.0	21
45	Oxolinic Acid	15.00	[M + H]^+^	262.1	216.0	29
					244.0	18
46	Ceftiofur	15.10	[M + H]^+^	524.0	241.0	20
					126.0	30
	Ceftiofur-d3	15.10	[M + H]^+^	527.0	244.1	15
	Metacycline	15.15	[M + H]^+^	443.1	426.2	16
47	Gamithromycin	15.40	[M + H]^+^	777.5	158.1	39
					619.7	32
48	Tilmicosin	15.70	[M + 2H]^++^	435.3	695.5	20
					174.1	30
49	Doxycycline	15.50	[M + H]^+^	445.2	428.1	10
					321.1	35
50	Nalidixic Acid^a^	16.80	[M + H]^+^	233.1	159.0	30
					187.0	25
51	Tiamulin	17.10	[M + H]^+^	494.3	192.1	20
					119.0	30
52	Penicillin G	17.15	[M + Na]^+^	357.1	198.1	20
					182.0	20
	Penicillin G-d7	17.15	[M + Na]^+^	364.0	205.2	13
53	Flumequine	17.20	[M + H]^+^	262.1	202.0	33
					244.0	19
54	Tylosina A	17.40	[M + H]^+^	916.5	174.1	36
					772.5	28
55	Erythromycin	17.60	[M + H]^+^	734.5	576.4	20
					158.1	30
56	3-O-Acetyltylosin	17.75	[M + H]^+^	958.5	174.0	36
					772.5	28
57	Oxacillin	18.20	[M + Na]^+^	424.1	265.1	20
					182.0	20
58	Penicillin V	18.20	[M + Na]^+^	373.1	182.0	20
					214.0	20
59	Cloxacillin	18.50	[M + Na]^+^	458.1	299.0	20
					182.0	20
60	Valnemulin	19.10	[M + H]^+^	565.4	263.1	20
					72.1	30
61	Dicloxacillin	19.20	[M + Na]^+^	492.0	333.0	20
					182.0	20
62	Nafcillin	19.30	[M + H]^+^	415.1	171.1	34
					199.1	13
63	Tilvalosin	19.45	[M + H]^+^	1042.6	174.0	39
					814.5	30
64	Rifaximin	20.60	[M + H]^+^	786.4	754.3	20
					736.3	30

^a^ Acquired only in milk.

**Table 3 molecules-24-02935-t003:** Precision, recovery and matrix effect (muscle and milk).

	Muscle				Milk			
Analyte ^a,b^	CV_r,pooled_ (%)	CV_Rw, pooled_ (%)	Rec (%)	ME ^c^ (%)	CV_r,pooled_ (%)	CV_Rw, pooled_ (%)	Rec (%)	ME ^c^ (%)
Sulfaguanidine	5.8	10	83	−14	5.0	14	61	**253**
Florfenicol Amine	3.2	6.1	85	**−22**	2.5	3.8	94	−13
Sulfanilamide	6.6	8.9	74	**−35**	5.8	12	66	**−21**
Desacetylcephapirin	7.6	7.3	76	−1	5.9	6.0	91	**−21**
Amoxicillin	4.5	6.3	64	−1	5.8	5.8	89	−10
Sulfadiazine	5.1	8.2	86	−18	4.8	8.6	74	−16
Sulfathiazole	5.2	8.0	83	−6	5.2	8.3	71	−15
Cephapirin	6.9	7.8	75	1	12	12	94	−5
Sulfapyridine	4.6	7.2	85	13	5.8	9.5	67	−13
Tildipirosin	6.2	10	72	−8	6.0	20	87	5
Cefquinome	6.1	7.2	97	−4	11	19	78	−16
Sulfamerazine	3.8	5.9	89	11	6.7	8.7	70	−13
Cefacetrile	12	13	80	**−26**	22	27	92	−5
Cefalonium	6.7	7.3	78	3	8.7	8.7	93	−6
Lincomycin	4.3	7.2	83	−12	2.3	4.4	92	**30**
Epitetracycline	6.2	9.2	66	**22**	4.7	7.6	96	5
Trimethoprim	3.4	7.8	90	−3	2.7	4.9	94	4
Thiamphenicol	10	11	87	−13	13	13	92	−13
Tulathromycin marker	5.6	8.6	81	−13	8.7	26	81	−18
Marbofloxacin	6.1	7.0	87	**−27**	4.3	4.9	96	−11
Sulfamethazine	4.1	7.2	85	−1	7.8	14	68	−13
Epioxytetracycline	8.8	13	62	**40**	13	15	90	−3
Norfloxacin	-	-	-	-	7.0	8.0	94	3
Tetracycline	6.3	9.4	71	**27**	4.5	5.0	91	**40**
Cefalexin	6.2	8.4	64	1	6.2	6.7	90	−7
Oxytetracycline	6.7	8.1	63	10	4.4	5.2	90	1
Ciprofloxacin	7.5	8.4	81	**−25**	6.1	7.5	95	−10
Enrofloxacin	4.7	6.6	93	−16	4.1	5.9	99	−1
Tulathromycin	8.3	16	69	10	7.5	13	94	5
Danofloxacin	5.7	7.2	90	−17	4.2	5.0	97	3
Cefazolin	8.4	9.0	83	−8	14	14	94	−11
Sulfamethoxazole	7.7	9.7	88	−16	6.7	6.7	84	**−24**
Difloxacin	5.0	7.4	93	−15	3.3	4.2	96	−13
Ampicillin	6.6	8.0	68	−2	9.4	10	88	−10
Sulfamonomethoxine	4.6	7.4	86	0	6.4	9.9	78	−19
Florfenicol	9.8	16	90	−17	13	13	94	**−28**
Cefoperazone	6.8	8.4	85	−13	16	18	103	**−25**
Sarafloxacin	5.8	7.4	86	**−26**	4.8	6.6	99	**−22**
Epichlorotetracycline	9.3	13	71	**65**	7.4	11	100	**22**
Neospiramycin	7.8	16	67	12	7.3	12	86	13
Chlortetracycline	5.8	7.7	69	**47**	5.7	7.2	92	**48**
Spiramycin	8.4	17	74	20	4.5	7.6	91	**21**
Sulfadimethoxine	4.5	8.1	88	−12	3.9	3.9	90	**−29**
Sulfaquinoxaline	6.1	7.7	86	**−26**	4.6	4.8	91	**−38**
Oxolinic Acid	4.9	6.7	97	4	2.6	6.6	96	−3
Ceftiofur	7.5	9.7	72	**−21**	6.7	7.1	95	**−24**
Gamithromycin	5.6	7.1	94	**55**	3.2	4.8	98	**26**
Tilmicosin	6.9	10	88	**49**	3.2	4.4	95	**38**
Doxycycline	7.0	9.5	69	1	6.6	7.4	95	10
Nalidixic Acid	-	-	-	-	3.6	6.0	94	−3
Penicillin G	7.9	9.0	85	**−21**	12	13	92	**37**
Tiamulin	9.3	13	88	**−21**	2.5	4.1	98	−5
Flumequine	4.3	7.3	95	−3	2.9	4.5	95	**52**
Tylosin A	7.7	15	85	**29**	2.5	4.8	96	**54**
Erythromycin A	4.9	8.6	89	**−27**	4.7	12	67	**−21**
3-O-Acetyltylosin	9.3	16	88	**42**	3.7	5.0	95	**53**
Oxacillin	6.0	11	83	**−29**	8.0	8.5	100	1
Penicillin V	7.9	11	85	**−28**	7.5	7.7	99	6
Cloxacillin	8.2	11	84	**−27**	9.8	12	110	**−25**
Valnemulin	19	31	75	**−26**	2.7	5.4	108	**55**
Dicloxacillin	6.7	10	81	**−25**	13	13	99	−10
Nafcillin	5.0	8.2	84	**−7**	5.0	5.4	99	−19
Tylvalosin	9.0	19	93	**50**	5.4	6.9	101	**90**
Rifaximin	9.0	12	91	**33**	4.8	9.6	98	7

^a^ Nalidixic acid and norfloxacin were not included in muscle method since these antibiotics were introduced later; ^b^ For valnemulin (muscle), cefacetrile (milk) and cefquinome (milk) the method can be used only for screening purposes (inadequate accuracy); ^c^ Values of matrix effect (ME) in bold are considered significant (>ǀ20ǀ%).

**Table 4 molecules-24-02935-t004:** Participation in Proficiency Test Schemes: comparison between methods.

Method	LC-Q-Orbitrap			LC-QqQ		
Sample Code/Year	Matrix	Analyte	Found Concentration (µg·kg^−1^)	Found Concentration (µg·kg^−1^)	Consensus Value (µg·kg^−1^)	Acceptability Range (µg·kg^−1^)
MI1432^-^A1/2014 ^a^	Milk	Sulfamethazine	144	96	103	57–150
MI1432-A2/2014 ^a^	Milk	Amoxicillin	5	ND	Not assigned	-
M1435-A1/2014 ^a^	Pig muscle	Sulfamethazine	88	75	69	36–102
	Pig muscle	Sulfadimethoxine	32	23	27	12–42
M1433-A2/2014 ^a^	Turkey muscle	Ciprofloxacin	5.5	5	5.6	3.2–8.1
	Turkey muscle	Enrofloxacin	173	152	160	92–227
MI1532-A1/2015 ^a^	Milk	Amoxicillin	16	ND	14	5.6–22
MI1532-A2/2015 ^a^	Milk	Sulfamethazine	165	131	134	75–191
MI1623-A1/2016 ^a^	Milk	Flumequine	91	111	88	47–129
MI1623-A2/2016 ^a^	Milk	Oxytetracycline	93	55	91	49–132
MI1715-A2/2017 ^a^	Milk	Danofloxacin	91	80	74	39–109
484 (material C)/2018 ^b^	Bovine Muscle	Marbofloxacin	178	193	170	100–240
484 (material C)/2018 ^b^	Bovine muscle	Oxytetracycline	89	79	106	59–152
334/2019 ^b^	Bovine muscle	Ciprofloxacin	10	10	NA^c^	-
334/2019 ^b^	Bovine muscle	Enrofloxacin	82	84	NA^c^	-
544/2019 ^b^	Bovine muscle	Tylosin A	54	87	NA^c^	-

^a^ Test Veritas, Padova, Italy; ^b^ RIKILT, Wageningen, Netherlands; ^c^ The final Report is not yet available.

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
