# Peer review of "Transfer of a Multiclass Method for over 60 Antibiotics in Food from High Resolution to Low Resolution Mass Spectrometry"

_molecules, 2019, doi:10.3390/molecules24162935_

Round 1

Reviewer 1 Report

Dear Authors,

thank you for submitting your manuscript to Molecules. The question of analyzing this complex mixtures is of great importance and your method might be useful. However, I have some questions and comments.

Page 12, line 184 ff. "However, with the low-resolution instrument, the more polar analytes, sulfanilamide, sulfaguanidine and florfenicol amine, were scarcely retained. This evidence wasascribed to the constitutive (hardware) features of the rather obsolete LC system coupled to triple quadrupole detector." This is very unfortunate formulation. The retention does not depend on hardware features but to the composition of the mobile phase, ion-pairing agents, and the stationary phase. Please remove or change the formulation.

Page 13, line 200 ff. " In this context, the ISs were not used for quantification purposes, but only to perform the internal quality control by checking the success of the analytical operations during the routine application of procedure" . How was then the quantitation performed? I was not able to find and clearly distinguis the description of that step so please add the description of methods used for quantitation.

page 24, line 403 ff. "In addition, examining LC-QqQ results, a series of accidental events (co-elution, selected precursor ions, fragmentation pathway etc) produced an overestimation of cefquinome in meat." I do not understand this statement. If you select the characteristic product ions in the qQq method originating from the mass of cefquinone which mechanism exactly causes the results to lead to a conclusion that the concentration is overestimated?

page 24, line 412 ff: "A further consideration must be done about the chromatographic conditions: the speed of the run should not be the analyst priority since also powerful MS detectors can fail when dozens of compounds are simultaneously determined in very complex matrices (false positive results and scarcer accuracy). Finally, since the here described method transfer has been performed on an obsolete LC-QqQ platform (fifteen  year-old), the feasibility of multiclass strategy was demonstrated also in absence of updated equipments." I understand what you wanted to say but this is quite a "rough" formulation, please rephrase.

Author Response

 Dear Authors,

thank you for submitting your manuscript to Molecules. The question of analyzing this complex mixtures is of great importance and your method might be useful. However, I have some questions and comments.

Page 12, line 184 ff. "However, with the low-resolution instrument, the more polar analytes, sulfanilamide, sulfaguanidine and florfenicol amine, were scarcely retained. This evidence wasascribed to the constitutive (hardware) features of the rather obsolete LC system coupled to triple quadrupole detector." This is very unfortunate formulation. The retention does not depend on hardware features but to the composition of the mobile phase, ion-pairing agents, and the stationary phase. Please remove or change the formulation.

Response. The formulation has been removed.

Page 13, line 200 ff. " In this context, the ISs were not used for quantification purposes, but only to perform the internal quality control by checking the success of the analytical operations during the routine application of procedure" . How was then the quantitation performed? I was not able to find and clearly distinguis the description of that step so please add the description of methods used for quantitation.

Response:

The quantification process was carried out using external standardization with matrix-matched curves and the subsequent correction of result for the recovery factor. This aspect has been more clearly explained in the revised manuscript. Generally, in the multiclass analysis of veterinary drugs labelled internal standards were not used for quantitative purposes since the costs and availability of a sufficient number of suitable (preferably corresponding) ISs prevent the application of isotopic dilution (internal standardization). Therefore, we used the ISs only for “qualitative” purposes monitoring the efficiency of the extraction process and the between-run differences in retention times. This approach is not uncommon and it is followed also by other researchers (see, e.g., the cited reference: Stolker et al. Comprehensive screening and quantification of veterinary drugs in milk using UPLC-ToF-MS. Anal. Bioanal. Chem. 2008, 391, 2309–2322)

page 24, line 403 ff. "In addition, examining LC-QqQ results, a series of accidental events (co-elution, selected precursor ions, fragmentation pathway etc) produced an overestimation of cefquinome in meat." I do not understand this statement. If you select the characteristic product ions in the qQq method originating from the mass of cefquinone which mechanism exactly causes the results to lead to a conclusion that the concentration is overestimated?

Response:

The statement is explained in detail at the beginning of paragraph 3.3. and I tried to explain better in the revised manuscript. Anyway, cefquinome and sulfamerazine shared the so-called “hidden” transitions. In particular, the transition m/z 265  > m/z 199 was monitored only for cefquinome (Table 2 of manuscript), but, although it was not monitored for sulfamerazine, this transition exists also for this latter (see Figure S6 – Supplementary Material). The co-elution of the analyte peaks together with the (accidental) selection of the same precursor ion (m/z 265) involve that cefquinome is overestimated if sulfamerazine is co-present, since sulfamerazine is able to increase the cefquinome signal.

page 24, line 412 ff: "A further consideration must be done about the chromatographic conditions: the speed of the run should not be the analyst priority since also powerful MS detectors can fail when dozens of compounds are simultaneously determined in very complex matrices (false positive results and scarcer accuracy). Finally, since the here described method transfer has been performed on an obsolete LC-QqQ platform (fifteen  year-old), the feasibility of multiclass strategy was demonstrated also in absence of updated equipments." I understand what you wanted to say but this is quite a "rough" formulation, please rephrase.

Response: The paragraph has been rephrased in the revised manuscript as follows: " A further consideration concerns the chromatographic separation which is fundamental also when highly selective detectors are applied. A good separation of peaks avoiding as many as possible the overlapping and bunching both of analytes and endogenous substances minimizes the risk of false positive results and reduces matrix effects. Finally, the here described method transfer has been successfully performed on an obsolete LC-QqQ platform (fifteen year-old), encouraging the implementation of multiclass strategy also in routine laboratories with limited instrumental resources”.

Reviewer 2 Report

TRANSFER OF A MULTICLASS METHOD FOR OVER 60 ANTIBIOTICS IN FOOD FROM HIGH RESOLUTION TO LOW RESOLUTION MASS SPECTROMETRY

In this manuscript the authors describe a multiclass method for screening and confirmatory analysis of antimicrobial residues in muscle and milk, which has been validated on two different platforms – LC-QqQ and LC-Orbitrap. The authors also address the critical issue of labs updating their platforms often and demonstrates that accurate and sensitive analysis can be performed on older instrumentation. I recommend this article be accepted for publications once the minor amendments below are addressed.

General comments and minor amendments

·         Line 120: The authors mention the storage time for sulphonamides was 24 months – is this class of compounds actually stable in solution for 24 months? Please comment. In addition, at what temperature were the stock standard solutions stored at?

·         Line 254: Linearity in matrix was calculated using only five point matrix matched curves – according to supplementary Table S3, the linearity range for both muscle and milk were mostly between 2-150 and 10-150 – which translates to only five or four point calibration curves. Could the authors provide some of calibration curves along with R2 and values and accuracy (%) data? Has a weighting been applied to the calibration curves?

·         Line 268: Table 3 – the authors report quite a few negative matrix effects – why is this the case?

 Author Response

Reviewer 2

In this manuscript the authors describe a multiclass method for screening and confirmatory analysis of antimicrobial residues in muscle and milk, which has been validated on two different platforms – LC-QqQ and LC-Orbitrap. The authors also address the critical issue of labs updating their platforms often and demonstrates that accurate and sensitive analysis can be performed on older instrumentation. I recommend this article be accepted for publications once the minor amendments below are addressed.

 General comments and minor amendments

Line 120: The authors mention the storage time for sulphonamides was 24 months – is this class of compounds actually stable in solution for 24 months? Please comment. In addition, at what temperature were the stock standard solutions stored at?

Response:

Thank you for this important comment. The stability and storage conditions of stock solutions is a very critical issue for development and daily management of confirmatory multiclass methods for veterinary drugs. Therefore, when in our laboratory we optimized the first multiclass method for antibiotics in meat (see the Tables reported in the Supplementary Material of our paper Moretti et al. Screening and confirmatory method for multiclass determination of 62 antibiotics in meat. J. Chromatogr. A 2016, 1429, 175–188), we also carried out an extensive study about the stability of analyte stock solutions. Sulphonamide stock solutions (100 µg/mL in methanol) demonstrated stability for at least 24 months at -20°C. This evidence was expected since the sulphonamide class is well known as highly stable. Other authors demonstrated, for example, the stability over (at least) a 12-months period both a +4°C and -18°C (see the cited references: Berendsen et al. Determination of the stability of antibiotics in matrix and reference solutions using a straightforward procedure applying mass spectrometric detection. Food Addit. Contam. - Part A Chem. Anal. Control. Expo. Risk Assess. 2011, 28, 1657-66; Gaugain et al. Stability study for 53 antibiotics in solution and in fortified biological matrixes by LC/MS/MS. J. AOAC Int. 2013, 96, 471-80). The storage temperature of stock solutions have been detailed in the revised manuscript.

Line 254: Linearity in matrix was calculated using only five point matrix matched curves – according to supplementary Table S3, the linearity range for both muscle and milk were mostly between 2-150 and 10-150 – which translates to only five or four point calibration curves. Could the authors provide some of calibration curves along with R2 and values and accuracy (%) data? Has a weighting been applied to the calibration curves?

Response:

The calibration curves were obtained applying unweighted least squared regression. The acquisition of five calibration points (six including the “zero”) was carried out following Commission Decision 2002/6757/EC (paragraph 3.1.1.5. of the Decision). It is worth to note that in this kind of methods (voluntary contamination), the quantification is a critical issue mainly when suspect samples must be re-prepared for confirmatory analysis in order to establish their compliance with the fixed MRLs. In these few cases, the confirmatory protocol describes the preparation of matrix-matched calibration standards encompassing the concentration of interest, i.e. the concentration estimated during the first (screening) analysis. As suggested by the Reviewer, some calibration curves were now reported in the Supplementary Material choosing some analytes belonging to different families and two interesting cases for macrolides (spiramycin and tulathromycin marker). Since R2 value is not a linearity index, we did not report it in the original manuscript since this approach can be criticized. It must be highlighted that Commission Decision 2002/657/EC does not include precise criteria to check linearity and, as explained in paragraph 3.2. of the manuscript, we applied SANTE/11813/2017 (“Guidance document on analytical quality control and validation procedures for pesticide residues analysis in food and feed”) in order to judge linearity. 

 Line 268: Table 3 – the authors report quite a few negative matrix effects – why is this the case?

Response:

Sorry, I not able to explain these results. Till today the reasons of matrix effect in ion source are not fully explained.

Reviewer 3 Report

The manuscript entitled “Transfer of a multiclass method for over 60 antibiotics in food from high resolution to low resolution mass spectrometry” showed the transfer of multiclass methods for antibiotics in meat and milk from LC-Q-Orbitrap (HR) to LC-QqQ (LR).

The study presents an interesting topic, the manuscript is well written and very clear. However, the article contains some weaknesses. The detailed comments are listed as follow:

The aim of this work should be more detailed, and the reason for this idea should be clarified in the introduction. There is a lack of aminoglycosides, by the way, a group of “difficult” analytes (especially if the method of sample preparation is based on the double extraction of one sample). A comparison of both techniques in the analysis of compounds containing this group would brought a lot of valuable information. “Standard solution”

There is no information about the storage temperature of stock solution. The informations presented in the following sentence are unclear: “Intermediate solutions (10, 1 and 0.1 μg mL-1) were prepared in H2O/ACN 75/25 (v/v) and in methanol for beta-lactams and for all other antibiotics, respectively”

The tables should be reorganized, they will be more readable if they are grouped into family types, the family name should also be added, and it will definitely make it easier for the reader to find the informations. For Figures S3, S6 and S7, the 0 and +15% lines should be bold or additionally marked, to improve the readability of the figures. Why is there no figure that shows the differences in precision between the LC-Q-Orbitrap and the LC-QqQ in muscles? “Comparison of LC-QqQ and LC-Q-Orbitrap methods”

In the paragraph presented above there is no table in which the weaknesses and strengths of the two compared techniques can be clearly seen, e. g.:

there is no information on the economic aspects of using both techniques the time taken to process the data ruggedness matrix effects

There is too much information about the cefquinome and sulfamerazine in this paragraph. 

Abstract, line 22: it should be “in the rage of”; page 21, line 330: it should be Ç€50Ç€%; page 22, line 370: Figure should be figure. Keywords should be in alphabetical order The presentation of references should be unified.

Author Response

Reviewer 3

The manuscript entitled “Transfer of a multiclass method for over 60 antibiotics in food from high resolution to low resolution mass spectrometry” showed the transfer of multiclass methods for antibiotics in meat and milk from LC-Q-Orbitrap (HR) to LC-QqQ (LR).

The study presents an interesting topic, the manuscript is well written and very clear. However, the article contains some weaknesses. The detailed comments are listed as follow:

The aim of this work should be more detailed, and the reason for this idea should be clarified in the introduction. There is a lack of aminoglycosides, by the way, a group of “difficult” analytes (especially if the method of sample preparation is based on the double extraction of one sample). A comparison of both techniques in the analysis of compounds containing this group would brought a lot of valuable information.

Response:

At the beginning of the “Results and discussion” Section (revised manuscript) an additional paragraph has been added explaining why these important antibiotics have been excluded.

“Standard solution”

There is no information about the storage temperature of stock solution. The informations presented in the following sentence are unclear: “Intermediate solutions (10, 1 and 0.1 μg mL-1) were prepared in H2O/ACN 75/25 (v/v) and in methanol for beta-lactams and for all other antibiotics, respectively”

Response:

Information about the storage temperature of stock solution has been added (-20°C). The unclear sentence has been rewritten. The stability and storage conditions of stock solutions is a very critical issue for development and daily management of confirmatory multiclass methods for veterinary drugs. Therefore, when in our laboratory we optimized the first multiclass method for antibiotics in meat (see the Tables reported in the Supplementary Material of our paper Moretti et al. Screening and confirmatory method for multiclass determination of 62 antibiotics in meat. J. Chromatogr. A 2016, 1429, 175–188), we also carried out an extensive study about the storage conditions of stock solutions

The tables should be reorganized, they will be more readable if they are grouped into family types, the family name should also be added, and it will definitely make it easier for the reader to find the informations. For Figures S3, S6 and S7, the 0 and +15% lines should be bold or additionally marked, to improve the readability of the figures. Why is there no figure that shows the differences in precision between the LC-Q-Orbitrap and the LC-QqQ in muscles?

Response:

The Figures S3, S6 and S7 (now S5, S7 and S9) have been modified introducing marked lines for +/-15% values, as suggested by the Reviewer. In addition, for the sake of completeness, a fourth figure (S8) has been added reporting the precision differences (LR vs HR) in muscle.

Sorry, I do not agree with the Reviewer about the table re-organization according antibiotic family. The current organization of Table 2 (MS parameters) follows the retention time order which is the criterion generally used. On the other hand, Table 3 follows the same order (retention times) and its re-organization will involve a discrepancy with the order (x-axis) of Figures S5, S7, S8 and S9 creating confusion. In my opinion the retention time order is more suitable. The reader can find the information about the family of each analyte in Tables S4 and S5.

“Comparison of LC-QqQ and LC-Q-Orbitrap methods”

In the paragraph presented above there is no table in which the weaknesses and strengths of the two compared techniques can be clearly seen, e. g.: there is no information on the economic aspects of using both techniques the time taken to process the data ruggedness matrix effects

Response:

I agree with the reviewer but the inclusion of a Table seems excessive. At the end of the paragraph 3.3., I summarized the main weakness and strengths of the two techniques. I discussed this comparison from the point of view of a routine laboratory since weakness and strengths are always related to the specific context. Of course, for a research group the HR system can be preferable.

There is too much information about the cefquinome and sulfamerazine in this paragraph. 

Response:

In the revised manuscript the information about cefquinome and sulfamerazine has been rewritten and shortened, as suggested by the Reviewer

Abstract, line 22: it should be “in the rage of”; page 21, line 330: it should be Ç€50Ç€%; page 22, line 370: Figure should be figure. Keywords should be in alphabetical order The presentation of references should be unified.

Response:

Corrected in the revised manuscript

Round 2

Reviewer 1 Report

Dear Authors,

thank you for revising the manuscript and provide explanation to my comments.

Please, check the phrasing once again and let a native English speaker or an experienced writer with sound knowledge of English to check the manuscript.

Kind regards